# Liraglutide Has Anti-Inflammatory and Anti-Amyloid Properties in Streptozotocin-Induced and 5xFAD Mouse Models of Alzheimer’s Disease

**DOI:** 10.3390/ijms22020860

**Published:** 2021-01-16

**Authors:** Leela Paladugu, Abeer Gharaibeh, Nivya Kolli, Cameron Learman, Tia C. Hall, Lixin Li, Julien Rossignol, Panchanan Maiti, Gary L. Dunbar

**Affiliations:** 1Field Neurosciences Institute Laboratory for Restorative Neurology, Central Michigan University, Mount Pleasant, MI 48859, USA; abeer.al-gharaibeh@iinn.com (A.G.); nivyareddy91@gmail.com (N.K.); learm1cr@cmich.edu (C.L.); hall2tc@cmich.edu (T.C.H.); rossi1j@cmich.edu (J.R.); maiti1p@cmich.edu (P.M.); 2Program in Neuroscience, Central Michigan University, Mount Pleasant, MI 48859, USA; 3Insight Research Center, Insight Institute of Neurosurgery and Neuroscience, Flint, MI 48607, USA; 4Physician Assistant Program, Central Michigan University, Mount Pleasant, MI 48859, USA; li6l@cmich.edu; 5College of Medicine, Central Michigan University, Mount Pleasant, MI 48859, USA; 6Field Neurosciences Institute, Ascension St. Mary’s, Saginaw, MI 48604, USA; 7Department of Psychology, Central Michigan University, Mount Pleasant, MI 48859, USA; 8College of Health and Human Services, Saginaw Valley State University, Saginaw, MI 48710, USA

**Keywords:** Alzheimer’s disease, liraglutide, streptozotocin, neuroinflammation, amyloid beta protein, sporadic Alzheimer’s disease, diabetes mellitus

## Abstract

Recent clinical and epidemiological studies support the contention that diabetes mellitus (DM) is a strong risk factor for the development of Alzheimer’s disease (AD). The use of insulin cell toxin, streptozotocin (STZ), when injected into the lateral ventricles, develops an insulin resistant brain state (IRBS) and represents a non-transgenic, or sporadic AD model (SAD), with several AD-like neuropathological features. The present study explored the effects of an anti-diabetic drug, liraglutide (LIR), in reversing major pathological hallmarks in the prodromal disease stage of both the 5xFAD transgenic and SAD mouse models of AD. Three-month-old 5xFAD and age-matched wild type mice were given a single intracerebroventricular (i.c.v) injection of STZ or vehicle (saline) and were subsequently treated with LIR, intraperitoneally (IP), once a day for 30 days. The extent of neurodegeneration, Aβ plaque load, and key proteins associated with the insulin signaling pathways were measured using Western blot and neuroinflammation (via immunohistological assays) in the cortical and hippocampal regions of the brain were assessed following a series of behavioral tests used to measure cognitive function after LIR or vehicle treatments. Our results indicated that STZ significantly increased neuroinflammation, Aβ plaque deposition and disrupted insulin signaling pathway, while 25 nmol/kg LIR, when injected IP, significantly decreased neuroinflammatory responses in both SAD and 5xFAD mice before significant cognitive changes were observed, suggesting LIR can reduce early neuropathology markers prior to the emergence of overt memory deficits. Our results indicate that LIR has neuroprotective effects and has the potential to serve as an anti-inflammatory and anti-amyloid prophylactic therapy in the prodromal stages of AD.

## 1. Introduction

Alzheimer’s disease (AD) is the most common, progressive, age-related neurological disorder. AD is characterized by impairment of memory and cognitive function, along with neurobehavioral abnormalities, and is the leading cause of death in the elderly population [1]. The hallmark pathology of this disease is the accumulation of amyloid beta (Aβ) proteins in the form of intraneuronal inclusions and extracellular neuritic plaques, intracellular hyperphosphorylated tau (pTau) in the form of neurofibrillary tangles (NFTs), along with synaptic loss and neuronal damage. AD is also associated with inflammatory responses, mitochondrial dysfunction, and oxidative stress. All the above-mentioned changes are prominent in the cortical and hippocampal regions of the brain [2,3,4]. Both genetic and environmental factors contribute to the pathogenesis of AD. A small percentage of cases of AD are associated with genetic factors, in the form of autosomal dominant inheritance of the genes producing amyloid precursor protein (APP), presenilin 1(PSEN1), and presenilin 2 (PSEN2), and is called the familial form (FAD) of AD [5], while the sporadic form (SAD) is associated with environmental risk factors, such as cardiovascular disease, stroke, cancer, impaired glucose tolerance, and diabetes mellitus (DM) [6,7,8].

An emerging body of research has demonstrated that DM is closely linked to AD. Impaired insulin signaling in the brains of AD patients is thought to play a pivotal role in establishing a link between both DM and AD [8,9]. Insulin signaling is initiated by the binding of insulin to its receptor on the cell membrane. This binding leads to the autophosphorylation of insulin receptor (IR) through tyrosine kinase activity. This, in turn, leads to further activation of downstream signaling proteins through phosphorylation of AKT, which further phosphorylates GSK-3β. Phosphorylated forms of GSK-3β inhibits the phosphorylation of tau and prevents the formation of Aβ oligomers and plaques [7,10,11].

Streptozotocin (STZ), a glucosamine-nitrosourea compound, obtained from a soil bacterium called *Streptococcus achromogenes*, is used as an experimental tool to develop animal models of type 2 DM (T2DM) [12]. STZ when injected intraperitoneally, damages the beta cells of the pancreas and causes insulin resistance. In a similar pattern, when STZ is injected via intracerebroventricular (i.c.v) route, it disrupts cerebral insulin signaling, energy metabolism, and promotes the accumulation of Aβ plaques, NFTs, and induction of progressive cognitive impairment. Therefore, STZ can be used to create animal models to study SAD pathology [13,14,15].

The established link between the two diseases motivated us to test whether repurposing anti- diabetic drugs would be an option to reduce the neuropathology in mouse models of AD. Previous studies have shown that incretin hormones, such as glucagon-like peptide-1 (GLP-1) analogues, like liraglutide (LIR), which are used in the treatment of DM, can have neuroprotective effects on AD brains by crossing the blood–brain barrier, acting as neurotrophic factors, and exerting neuroprotective effects, such neurite growth and anti-inflammation [16].

In addition, the neuroprotective effects of LIR in intermediate and late stages of AD have been documented [17,18]. However, there is no definitive evidence that LIR can prevent AD-like pathology in the early stages of the disease. Although the definitive diagnosis of AD was originally made only through autopsy [19,20], pathological changes in the brain can happen long before behavioral symptoms appear. To reflect these changes, the National Institute of Aging (NIA) has designated three stages of AD: Preclinical, mild cognitive impairment (MCI), and dementia. The NIA pre-clinical or prodromal stage refers to pathophysiological changes in the brain that occur prior to, or during, the initial stages of memory loss [21,22].

To address whether LIR has neuroprotective effects in the prodromal stage of AD, we used both wild type (WT) mice with STZ injections as SAD model and young transgenic 5xFAD mice to investigate and compare the neuroprotective effects of LIR on neurodegeneration, neuroinflammation, Aβ-plaque deposition and insulin-signaling changes in the brain, as well as assessing the level of cognitive function in both models of prodromal AD.

## 2. Results

Because no significant gender differences were observed on any of the measures in this study, the males and females were combined within each of the 8 sub-groups. All mice survived all surgical and treatment interventions, so the analyses were performed on all the mice in the study.

### 2.1. No Between-Group Differences Were Observed on Any Behavioral Measures

Analyses of passive avoidance (PA), open-field (OF), and object-recognition (NOR) tasks are represented in Figure 1. To test the differences in learning and memory on PA task, the ability of the mice to avoid an aversive stimulus, as measured in latency to enter the shock chamber, was analyzed by repeated-measures ANOVA. There were no significant between-group differences in the latency to avoid the shock in all trials [*F*
_(7,47)_ = 2.02, *p* = 0.07; Figure 1E,F]. There were also no significant between-group differences in the total time spent in the periphery (Figure 1A) and in the center (Figure 1B) of the OF over 60 min and in the number of entries into the familiar object (Figure 1C) and novel-object (Figure 1D) zone in the NOR task.

### 2.2. No Between-Group Differences Were Observed in the Number of Degenerating Neurons

One-way ANOVA on the average number of fluorojade-B-(FJB) labeled degenerated cells in the cortical, CA1 and CA3 regions of the brain revealed no significant differences between groups (data not shown).

### 2.3. LIR Reduced Activation of Astrocytes in the Cortical and Hippocampal CA1 and CA3 Regions of SAD and 5xFAD Mice

One-way ANOVAs on the average number of GFAP immunoreactive (GFAP-IR) cells in the cortical region of the brain (Figure 2A,B) showed an overall significant difference between groups [*F*
_(7,122)_ = 21.03, *p* < 0.05]. Tukey’s post hoc tests indicated that there was significant difference in the number of GFAP-IR cells in the WT + Vehicle group when compared to the WT+STZ group (*p* < 0.001), with the latter showing increased numbers of activated astrocytes. LIR treatments significantly reduced the number of activated astrocytes in the WT + STZ + LIR and WT + LIR groups when compared to the WT + STZ group (*p* < 0.001). There were no between-group differences in the number of GFAP-IR cells in the 5xFAD mice. However, significant increases in the GFAP-IR cells were seen in the vehicle and STZ-treated groups and between WT and 5xFAD mice (*p* < 0.001), with 5xFAD + Vehicle and WT + STZ groups showing increased numbers of GFAP-IR cells. There was a significant increase in the number of GFAP-IR in the vehicle and STZ-treated groups of 5xFAD mice, when compared to WT mice (*p* < 0.001). There were no significant differences in the number of GFAP-IR cells in the STZ + LIR and LIR groups of 5xFAD mice, when compared to WT mice (*p* = 0.723 and *p* = 0.312, respectively; Figure 2A,B).

One-way ANOVAs for the average number of GFAP-IR cells in the CA1 hippocampal region (Figure 2C,D) showed significant difference between groups in both WT and 5xFAD mice [*F*
_(7,136)_ = 57.806, *p* < 0.05]. Tukey’s post hoc tests revealed a significant difference in the WT + Vehicle group, when compared to WT + STZ group (*p* < 0.001) with the later showing increased number of astrocytes. LIR treatment significantly reduced the number of astrocytes in the WT + STZ + LIR and WT + LIR groups, when compared to WT + STZ group (*p* < 0.001). In 5xFAD mice, there was a significant decrease in the number of GFAP-IR cells in the 5xFAD + STZ + LIR mice when compared to 5xFAD + STZ group. There was a significant decrease in the number of GFAP-IR cells in the 5xFAD + LIR mice when compared to the 5xFAD + Vehicle mice (*p* < 0.01) Also, there was a significant increase in the number of GFAP-IR cells in the 5xFAD + Vehicle, compared to WT + Vehicle groups (*p* < 0.001). There were no significant differences between 5xFAD + STZ and WT + STZ groups (*p* = 0.08) and between 5xFAD + STZ + LIR and WT + STZ + LIR groups (*p* = 0.57). The number of GFAP-IR cells in WT + LIR was significantly lower when compared to 5xFAD + LIR group (*p* = 0.05; Figure 2C,D). The same pattern of expression was observed in CA3 [*F*
_(7,124)_ = 28.60, *p* < 0.05] region of the brain, as shown in the CA1 region (Figure 2E,F).

### 2.4. LIR Reduced Microglial Activity in the Cortical and Hippocampal CA1 and CA3 Regions in SAD and 5XFAD Mice

One-way ANOVAs on the average number of Iba-1 positive cells in the cortical region of the brain showed an overall significant difference between groups in both WT and 5xFAD mice [*F*
_(7,144)_ = 60.298, *p* < 0.05]. Tukey’s post hoc tests revealed a significant difference in the WT + Vehicle group when compared to WT + STZ group (*p* < 0.001), with the latter showing increased number of microglia (Figure 3A,B). LIR treatment significantly reduced the number of microglia in the WT + STZ + LIR and WT + LIR groups when compared to WT + STZ group (*p* < 0.001). There was a significant increase in the number of Iba-1 positive cells in STZ group when compared to vehicle group in 5xFAD mice (*p* < 0.001). LIR treatment significantly reduced the number of microglia in the STZ + LIR and LIR groups when compared to STZ group in 5xFAD mice (*p* < 0.001). There was a significant increase in the number of Iba-1 positive cells in the 5xFAD + Vehicle group when compared to WT + Vehicle group (*p* < 0.001). The number of Iba-1-positive cells in WT + LIR is significantly lower when compared to 5xFAD + LIR group (*p* = 0.000; Figure 3A,B).

One-way ANOVAs on the average number of Iba-1-positive cells in the CA1 (Figure 3C,D) and CA3 (Figure 3E,F) hippocampal regions of the brain showed significant differences between groups in both SAD and 5xFAD mice [*F*
_(7,151)_ = 90.51, *p* < 0.05 and *F*
_(7,157)_ = 89.74, *p* < 0.05, respectively]. Tukey’s post hoc tests revealed the same pattern of expression in the number of Iba-1 cells, as shown in the cortical region of the brain, except no significant difference in the number of Iba-1-positive cells between WT + STZ and 5xFAD + STZ was observed in the CA1 and CA3 regions of the brain.

### 2.5. LIR Reduced the Amount of Aβ Levels in the Cortical and the Hippocampal Regions of the 5xFAD Mice, but not in SAD Mice

Immunoreactivity with 6E10 antibody revealed that Aβ protein predominantly formed intraneuronal inclusions, with very few extracellular plaques. Extracellular plaques were observed in the cortical areas of WT + STZ and 5xFAD + Vehicle groups (Figure 4A), while CA1 and CA3 regions, in both models, had prominent intraneuronal inclusions (Figure 4D,G). One-way ANOVAs of the Western blot bands indicated an overall significant difference in the levels of Aβ between groups in the cortical regions of SAD and 5xFAD mice [*F*
_(7,24)_ = 362.34, *p* < 0.05]. There were no significant differences in the levels of Aβ between the vehicle-treated WT mice, when compared to WT + STZ, WT + STZ + LIR, or WT + LIR (*p* = 0.99, *p* = 0.87, and *p* = 0.28, respectively). Surprisingly, LIR increased the expression of Aβ levels in the WT + LIR group when compared to the WT + STZ group (*p* = 0.01), although the Aβ levels in all four of these groups were very low in the cortex. The level of cortical Aβ levels was significantly elevated in 5xFAD + Vehicle group when compared to all other groups (*p* < 0.001). LIR treatment significantly reduced the levels of Aβ in the 5xFAD + LIR group when compared to the 5xFAD + Vehicle group (*p* < 0.001). The 5xFAD mice had significantly increased cortical levels of Aβ in comparison to WT mice in both the vehicle and STZ groups (*p* < 0.001), with LIR treatment significantly reducing the levels of Aβ in the 5xFAD + LIR group when compared to WT + LIR group (*p* < 0.001) (Figure 4B,C).

One-way ANOVAs revealed an overall significance in the hippocampal Aβ levels, between groups in WT and 5xFAD mice [*F*
_(7,24)_ = 543.67, *p* < 0.05]. Tukey’s post hoc tests revealed a significant increase in the levels of hippocampal Aβ levels in WT + STZ group and WT + STZ + LIR groups when compared to WT + Vehicle group (*p* < 0.001) following STZ injections. LIR treatment significantly decreased the expression of hippocampal Aβ levels in the WT + LIR group when compared to WT + STZ + LIR group (*p* < 0.001). However, the Aβ levels were slightly increased in the WT + STZ + LIR group when compared to the WT + STZ group but the changes were not significant. The hippocampal Aβ levels was increased significantly in the 5xFAD + Vehicle group when compared to all the other groups (*p* < 0.001). LIR treatment significantly reduced the levels of hippocampal Aβ levels in STZ + LIR and LIR group when compared to the vehicle group in 5xFAD mice (*p* < 0.001). The 5xFAD vehicle-treated mice had significantly increased Aβ levels in comparison to all groups of WT mice and all other groups of 5xFAD mice (*p* < 0.001, in all cases). As was the case in the cortex, the STZ treatments prevented much of the increase in Aβ levels in the hippocampus. Importantly LIR treatments significantly reduced the Aβ levels in the 5xFAD + STZ + LIR- (*p* < 0.001) and 5xFAD + LIR group (*p* < 0.01) when compared to the same groups in WT mice (*p* < 0.001; Figure 4E,F). Results of the Western blot analyses of Aβ levels mirrored those observed in the immunohistochemical studies (Figure 4A,D,G,H).

### 2.6. Liraglutide Increased the Levels of Insulin-Degrading Enzyme (IDE) in the Cortical and Hippocampal Regions of WT and 5xFAD Mice, While Liraglutide Increased Phosphorylated Insulin Receptor (pIR) in the Cortical but Not the Hippocampal Regions of the WT and 5xFAD Mice

Overall significant differences were observed in IDE levels in both cortical [*F*
_(7,24)_ = 82.55, *p* < 0.05] and hippocampal [*F*
_(7,24)_ = 15.079, *p* < 0.05] regions of the brain (Figure 5A–D). Tukey’s post hoc tests revealed a significant decrease in IDE levels in the cortical regions of the brain in the WT + STZ group when compared to WT + Vehicle group (*p* < 0.001). LIR treatments prevented the decrease of IDE in the WT + STZ + LIR and WT + LIR groups, when compared to WT + Vehicle and WT + STZ mice (Figure 5A,C). No significant differences were observed in IDE levels for the 5xFAD + Vehicle when compared to 5xFAD + STZ and 5xFAD + STZ + LIR mice (*p* = 0.23 and *p* = 0.89, respectively). LIR treatments prevented the decrease in IDE levels in the 5xFAD + LIR group when compared to 5xFAD + Vehicle and 5xFAD + STZ + LIR groups (*p* < 0.01).

There was a significant decrease in IDE levels in the WT + STZ group when compared to the WT + Vehicle group in WT mice in the hippocampus (Figure 5B,D). LIR treatments prevented the loss of IDE in the STZ + LIR when compared to the vehicle-treated and STZ-treated groups of WT mice. There was no significant difference of IDE levels in the 5xFAD + STZ and 5xFAD + STZ + LIR groups when compared to the 5xFAD + Vehicle group. LIR treatment prevented loss of IDE in LIR group when compared to the remaining groups in 5xFAD mice (*p* < 0.01).

There were no significant differences in the levels of IDE in 5xFAD + Vehicle mice when compared to their WT + Vehicle counterpart in the cortical region of the brain (Figure 5C). There was a significant increase in the levels of IDE in the 5xFAD + STZ mice when compared to WT + STZ mice (*p* = 0.043). Following LIR treatment, there were significant differences in the levels of IDE in the 5xFAD + STZ + LIR and 5xFAD + LIR mice when compared the WT + STZ + LIR and WT + LIR mice (*p* < 0.01, Figure 5C). In the hippocampus, there was no significant difference in the levels of IDE in the 5xFAD + Vehicle and the 5xFAD + LIR groups when compared to their WT counterparts (Figure 5D). There was a significant decrease in the levels of IDE in the WT + STZ mice when compared to their 5xFAD counterparts. Following LIR treatment, there were significant differences in the levels of IDE in the 5xFAD + STZ + LIR and the 5xFAD + LIR mice when compared to the same groups of WT mice (Figure 5D).

One-way ANOVAs revealed significant differences in the levels pIR in the cortical [*F*
_(7,24)_ = 30.816, *p* < 0.05] and hippocampal [*F*
_(7,24)_ = 7.111, *p* < 0.05] regions of the brain (Figure 5A,B,E,F). Tukey’s post hoc tests revealed a significant decrease (*p* = 0.019) in the level of cortical pIR in the WT + STZ mice when compared to WT + Vehicle mice (Figure 5E). There was no significant treatment effect (*p* = 1.00) of LIR in WT + STZ + LIR mice when compared to WT + STZ mice. There were no significant differences in the level of cortical pIR between the 5xFAD + Vehicle and 5xFAD + STZ mice (*p* = 1.00). However, LIR treatment significantly increased the cortical pIR levels in 5xFAD + STZ + LIR (*p* < 0.01) and 5xFAD + LIR (*p* < 0.01) groups when compared to 5xFAD + Vehicle mice (Figure 5E).

As shown in Figure 5B,F, there was no significant difference in the levels of hippocampal pIR between the WT + Vehicle and WT + STZ mice (*p* = 0.98). Although we see a trend of gradual increase in the hippocampal pIR in the WT + STZ + LIR and WT + LIR groups, the difference was not significant (*p* = 0.99). No significant difference was observed in the levels of hippocampal pIR in 5xFAD + Vehicle mice when compared to 5xFAD + STZ mice (*p* = 0.55). However, LIR treatments significantly reduced the levels of hippocampal pIR in the 5xFAD + STZ + LIR, and 5xFAD + LIR mice when compared to 5xFAD + Vehicle mice (*p* < 0.01).

There was a significant reduction in the levels of pIR in the 5xFAD + Vehicle group when compared to WT + Vehicle (*p* = 0.011), and a significant increase in the levels of pIR in the 5xFAD + STZ + LIR (*p* < 0.01) and the 5xFAD + LIR (*p* < 0.01) groups when compared to their WT counterparts in the cortical region of the brain (Figure 5E). There was a significant increase in the levels of pIR in the vehicle-treated 5xFAD + Vehicle mice when compared to the WT + Vehicle mice, but no significant differences were observed in the levels of pIR in the 5xFAD + STZ + LIR, and LIR groups in 5xFAD mice when compared to their WT counterparts in the hippocampus (Figure 5F).

### 2.7. Liraglutide Did Not Increase the Levels of pAKT in the Cortical and the Hippocampal Regions of the Brain in SAD and 5xFAD Mice, While Liraglutide Increased the Levels of pGSK3β in the Cortical Region of both SAD and 5xFAD Mice and Only in the Hippocampal Region of 5xFAD Mice

Significant differences were found in the levels of pAKT in both cortical [*F*
_(7,24)_ = 44.797, *p* < 0.05] and hippocampal [*F*
_(7,24)_ = 8.643, *p* < 0.05] regions of the brain (Figure 6A–D). In the cortical region, LIR treatment significantly increased the levels of pAKT in the WT + LIR group when compared to all the other groups, while no significant differences were found between the WT + Vehicle, WT + STZ and WT + STZ + LIR mice (Figure 6C). However, in the 5xFAD + Vehicle mice, higher levels of cortical pAKT were observed, when compared to all other groups (*p* < 0.01). LIR treatment reduced the levels of cortical pAKT in the WT + LIR group when compared to all other groups (Figure 6A,C).

In the hippocampal region, there were no significant differences in the levels of pAKT between all the WT groups (Figure 6D). However, LIR treatment significantly increased the levels of hippocampal pAKT in the 5xFAD + STZ + LIR group when compared to the 5xFAD + STZ group (*p* = 0.006). However, there was a significant decrease in the levels of pAKT in the 5xFAD + LIR group when compared to the 5xFAD + STZ + LIR group (*p* < 0.05; Figure 6B,D).

There was a significant increase in the levels of cortical pAKT in vehicle-, STZ-, and STZ + LIR-treated 5xFAD mice when compared to their WT counterparts (Figure 6C). There was also significant increase in the levels of hippocampal pAKT in vehicle-, and STZ + LIR-treated groups in 5xFAD mice when compared to their WT counterparts (Figure 6D).

Significant differences were found in the levels of pGSK3β in both cortical [*F*
_(7,24)_ = 213.963, *p* < 0.05] and hippocampal [*F*
_(7,24)_ = 48.321, *p* < 0.05] regions of the brain (Figure 6E,F). Tukey’s post hoc tests revealed significant difference in cortical pGSK3β levels in the WT + STZ + LIR and WT + LIR groups when compared to WT + Vehicle and WT + STZ mice (*p* < 0.01; Figure 6A,E). Tukey’s post hoc tests also revealed a significant increase in the levels of cortical pGSK3β in 5xFAD + LIR mice when compared to all the other 5xFAD mice (*p* < 0.01; Figure 6A,E).

Hippocampal pGSK3β levels was significantly higher (*p* < 0.01) for the WT + Vehicle mice than all other groups and LIR treatments did not affect the levels of pGSK3β (Figure 6F). No significant differences between the 5xFAD + Vehicle and 5xFAD + STZ groups (*p* = 0.894) were observed, but LIR treatments significantly increased the levels of hippocampal pGSK3β in the 5xFAD + STZ + LIR group when compared to 5xFAD + Vehicle mice and in the 5xFAD + LIR group when compared to the 5xFAD + STZ group (Figure 6B,F).

There was a significant decrease in the levels of cortical pGSK3β in 5xFAD + STZ + LIR and 5xFAD + LIR groups when compared to the same groups in WT mice (Figure 6E). There was also a significant decrease in the levels of hippocampal pGSK3β in the hippocampal region of 5xFAD + Vehicle mice when compared to the WT + Vehicle mice, but a significant increase in the levels of hippocampal pGSK3β was found in 5xFAD + LIR mice when compared to their WT + LIR counterparts (Figure 6F).

## 3. Discussion

The aim of this study was to assess and compare whether LIR would have protective effects on prodromal neuropathology in sporadic and 5xFAD mouse models of AD. In our study we injected STZ into 3-month-old WT mice to mimic sporadic AD [13,15] and used 3-month-old 5xFAD mice that were given STZ or vehicle injections to assess the differences in the expression of Aβ pathology, neurodegeneration, neuroinflammation and insulin-signaling pathways. In addition, we further evaluated whether treatments of the GLP-1 analogue, LIR, could reduce those deficits. We observed significant increased neuroinflammation in the cortical and hippocampal regions of the brain in both SAD and 5xFAD mice, along with Aβ pathology and STZ-induced disruption of the insulin-signaling pathway, and that treatment with LIR ameliorated these changes. Previous studies demonstrated neuroprotective effects after LIR treatment in 7–9 month- and 14–16 month- old APP/PS1 mice [17,18]. Although, studies have been done to check the prophylactic effects of long-term administration of LIR in two-month-old double transgenic mice [23], to the best of our knowledge, this is the first study to assess the neuroprotective effects of LIR in the brains of both sporadic and transgenic 5xFAD mouse models that mimic the prodromal stage of the AD.

Accumulation of Aβ protein in the form of plaques is a major pathophysiological finding in AD. The deposition of Aβ aggregates can trigger neurodegeneration, which can lead to cognitive disorders [24]. Our study was designed to assess whether LIR treatments could alleviate Aβ pathology prior to the onset of cognitive decline. We observed no significant differences between groups, on behavioral measures of open-field activity levels, recognition memory on novel-object recognition test and episodic memory as determined by increased latency to enter in the area where shocks were previously given during passive-avoidance tasks. This outcome indicates that these mice accurately represented the prodromal stages of AD.

Previous studies have shown that cognitive deficits in 5xFAD mice start to emerge at 4–5 months of age, therefore we didn’t expect to see any cognitive deficits at 3 months of age in 5xFAD mice [25,26,27] but our behavioral analyses was completed when the mice reached 4.7 months of age, just around the time where cognitive deficits emerged in these animals. We think different testing parameters between studies may be the reason for discrepant findings between studies. The most salient difference may be that the repeated testing conducted in our study, especially with PA task, at this early age may have protected against deficits that would have otherwise emerged with time. As such, our results are consistent with studies which reported that the inflammation and neuropathology, in the form of Aβ plaques and tau tangles, start to accumulate about a decade or more before the actual cognitive problems begin in AD patients [24,28]. We found that following STZ injections, there was increased accumulation of intraneuronal Aβ inclusions along with a few extracellular Aβ plaque deposits in both SAD and 5xFAD mouse models. Although we did not count the extra cellular plaques, we noticed very few extracellular plaques in few cortical sections of WT + STZ and 5xFAD + Vehicle mice. An interesting finding, we noticed in our study was that STZ injections did not increase the Aβ levels in 5xFAD + STZ group and LIR seems to have no effect on the levels of Aβ in 5xFAD + STZ + LIR group, as evidenced by both immunohistochemical and Western blot analyses. However, there was significant decrease in the levels of Aβ in the 5xFAD + LIR group when compared to 5xFAD + Vehicle, showing a treatment effect of LIR. We think this is an important finding in this study, which eventually validated the use of LIR. This observation warrants further detailed study in order to discern the role of STZ in transgenic mice. We also observed no significant difference in the levels of Aβ in the WT + STZ + LIR group when compared to WT + STZ group. This might be due to the relatively lower levels of Aβ and that LIR did not have enough of Aβ to show a significant effect. However, we assume that as the animal ages and following long term administration of LIR, there might be a significant treatment effect. In accordance to results from previous studies, which showed amelioration of Aβ pathology in transgenic mouse models [29], LIR treatment reduced the Aβ levels 5xFAD mice. The accumulation of Aβ inclusions and few plaques in the present study may be due to STZ-induced formation of reactive oxygen species (ROS). In aging and neurodegeneration, there is a decline in the antioxidative mechanisms and previous studies have reported the ability of ROS to modify crucial molecules and proteins responsible for neurodegenerative diseases, like AD [30,31]. LIR has the potential to serve as an antioxidant, as evidenced by in vitro studies done in SH-SY5Y cells [32,33] and in in vivo studies in 3xTg female mice [34]. Although ROS levels were not directly assessed in our study, we hypothesize that LIR over time, can ameliorate the STZ-induced changes in Aβ of SAD and 5xFAD mice by an antioxidant effect, a goal for future study in this area.

An increase in neuroinflammation, along with activation of astrocytes and microglia, is one of the prominent pathologies observed in AD brain [35,36]. A very early inflammatory response, through release of potent pro-inflammatory markers and activated microglia, happens before Aβ accumulation and this affects the clearance of Aβ plaques [37,38,39]. In agreement with previous studies [40,41,42], we observed an increased number of astrocytes and microglia in the cortical, and the hippocampal (CA1 and CA3) regions of the brain following STZ injections, with changes more pronounced in the SAD mice than in 5xFAD mice. Alleviation of these deficits with 30 days of LIR treatment in our study, correspond to findings by others who suggest that LIR was shown to have anti-inflammatory action in a double transgenic mouse model of AD [17], through cAMP regulation of MAPK and JNK pathways [23,43,44] and by modulating insulin receptors [29,45].

Our findings are consistent with previous studies which reported that the most obvious and important alteration in the brain that happens following administration of STZ is neuroinflammation and that it is more pronounced in SAD than transgenic AD model [46]. It is evident from previous studies that DM is associated with inflammation, and a risk factor for the development of SAD. Because LIR is used to treat DM, its use as an anti-inflammatory agent for treating AD, especially SAD seems warranted. Our results provide further evidence that LIR can be considered a promising anti-inflammatory agent in AD, even when used in the prodromal stages of the disease.

Recent studies have shown that incretin hormones, like LIR, are effective in restoring insulin sensitivity in AD mouse models [29,45]. The neuroprotective and neurotropic roles of LIR in AD mouse models have been under consideration for treating neurodegenerative diseases, like AD [47]. Insulin signaling in the brain occurs mainly via two phosphorylation pathways, PI3K/AKT and mitogen-activated protein kinase (MAPK) pathway [9,48,49]. Our study focused on pathway where AKT and GSK3 activity regulates phosphorylation of tau and mediates Aβ neurotoxicity. Although 5xFAD mice do not express neurofibrillary tangles, results from our study demonstrated that changes in the pIR, AKT and GSK3 expression correspond to disrupted insulin-signaling pathway in the cortical and hippocampal regions of the brain in both SAD and 5xFAD models.

Insulin-degrading enzyme, as the name suggests, degrades excess insulin in the body and plays a key role in degrading Aβ oligomers in vitro and in vivo under normal circumstances [50]. STZ injections reduce the expression of IDE and contribute to accumulation of Aβ oligomers in the brain [51,52,53]. Inability of the body to clear Aβ oligomers and increased microglial activity are thought to reduce the expression of IR and induce re-distribution and internalization of insulin receptors on the cell surface [48,54,55]. This reaction downregulates the downstream signaling of AKT and other major proteins, finally, contributing to Aβ neurotoxicity [48,54]. In agreement with results from the above-mentioned studies, our study demonstrated reduced expression of IDE and pIR in both mouse models of AD following STZ injections, but significantly more pronounced in SAD than in 5xFAD mice. LIR restored the expression of IDE in the cortical and hippocampal regions of the brain. These results are consistent with previous studies which showed that STZ injections reduced IDE expression to a larger extent in SAD than in transgenic mouse models [47]. LIR also restored the levels of pIR in the cortical region of 5xFAD mice while it could just restore the levels of pIR in the hippocampal region of SAD mice but not in 5xFAD mice. In fact, LIR further reduced the levels of pIR in the 5xFAD mice after STZ injection. We noticed a significant increase in expression of cortical pIR in the 5xFAD + LIR group when compared to 5xFAD + Vehicle group. However, the same pattern was not observed in the hippocampal region of 5xFAD mice.

We also observed a reduced expression of pAKT in both the cortical and hippocampal regions in both AD models following STZ injections, but treatments with LIR did not significantly elevate the pAKT levels in the SAD mice (except in the WT + LIR group) and in 5xFAD mice; in fact, LIR reduced the pAKT levels in 5xFAD + LIR group when compared to 5xFAD + Vehicle group in both cortical and hippocampal regions of the brain. We noticed a trend towards LIR causing a further reduction of cortical pAKT levels in the WT + STZ + LIR mice when compared to WT + STZ group but the changes were not significant. Overall, our results suggest that expressed levels of pAKT and pIR are difficult to predict in these relatively young mice. Since the mice were only 4.7 months of age at the time of analyses, the disruption of PI3K/AKT pathway may not have yet reached its peak. The changes of pIR and pAKT we observed, reflect the very early stages of disruption, although an initial surge in the expression of these proteins may have stabilized as the disease progressed. With respect to the expression of pGSK3β, we noticed LIR treatment significantly restored the levels of pGSK3β in both cortical and hippocampal regions of the brain in both mouse models of AD, but to a greater extent in the SAD model.

Overall, our results provide in vivo evidence that the initial stages of impaired insulin signaling, especially in 5xFAD mice brain can cause increased Aβ protein levels through the activity of IDE, along with increased inflammatory response, to a greater extent in SAD mice, which is in keeping with evidence from previous studies [50,56]. Our results also confirm that LIR has significant effects on the insulin signaling pathway, by the increasing the expression of IDE, pIR, and pGSK3β, which may be related to improvement of insulin signaling, the key downstream signaling proteins which are thought to be neuroprotective for the cell [57]. Since we did not find any significant evidence for degenerating neurons, but found intraneuronal Aβ inclusions and prominent neuroinflammation, our results support the findings from previous studies which have shown that insulin-signaling dysfunction in the AD brain is correlated to neurodegeneration and Aβ pathology [11,58].

When we compared the differences in the neuropathology induced by injections of STZ in both SAD and 5xFAD models, we noticed that neuroinflammation is an immediate and significant response to STZ in SAD mouse model than 5xFAD model, while Aβ pathology had a significant response to STZ in the 5xFAD mouse model than in the SAD model, along with disruption of insulin signaling markers like IDE, pIR, and pGSK3β, to a greater extent in SAD mice. This pathology was significantly reversed by administration of LIR. As such, our results strongly support the hypothesis that early intervention with anti-diabetic drugs might prove beneficial in reversing the neuroinflammatory pathology in SAD and amyloid pathology in 5xFAD mice in their prodromal stage of AD. Further research is required to assess the long-term consequences of early LIR intervention and to compare its efficacy when treatments are delayed.

## 4. Materials and Methods

### 4.1. Chemicals

All sources for the chemicals and antibodies used in this study are shown in Appendix A.

### 4.2. Animals

For this study, we used 24, three-month-old 5xFAD and 24 age-matched wild type (WT) mice. The mice from these two groups were randomly assigned into the following 8 sub-groups (*n* = 6), with approximately 50% being male and 50% being female mice in each group: WT + Vehicle, WT + STZ, WT + STZ + LIR, WT + LIR, 5xFAD + Vehicle, 5xFAD + STZ, 5xFAD + STZ + LIR and 5xFAD + LIR. All the mice were housed in the vivarium in polyethylene bins on a continuous 12-h day/night cycle. Mice had access to water and food, ad libitum, and were kept at the same conditions of temperature and humidity. This study was conducted in accordance with the protocols approved by the Institutional Animal Care and Use and Committee of the Central Michigan University (protocol # 15–20; 16 July 2015). A schematic representation of the experimental design and treatment timeline of the study was represented in Figure 7.

### 4.3. Animal Surgeries

Surgeries using a stereotaxic device (Kopf Instruments, Tujunga, CA, USA) were performed in all the mice at 3 months of age as described previously [44]. Mice were anesthetized using 2% isoflurane, combined with oxygen at 0.8 L/min throughout the procedure. The mice were monitored continuously throughout surgery, and isoflurane and oxygen supply were adjusted as needed to maintain an appropriate level of anesthesia. Prior to surgery the mice were cleaned with alcohol and the area of the incision was shaved. The mice were stabilized with ear bars on the stereotaxic device, to keep lambda and bregma on the same plane. The surgical site on the head was cleaned with chlorhexidine (Molnycke Healthcare, Norcross, GA, USA). A midline incision was made on the scalp, fascia was retracted, and two burr holes were made through the skull directly above the lateral ventricles on both sides (coordinates relative to bregma: anterior + 0.5 mm; medial–lateral ± 1.1 mm; dorsal-ventral ± 2.0 mm). The saline or STZ (3 mg/kg) solution was loaded into a 10-μL Hamilton micro-syringe and each mouse received bilateral injections of either STZ or vehicle at a constant rate of 0.3 μL/min. After a 3-min diffusion period, the syringe was withdrawn and re-positioned into the contralateral hemisphere and the injection procedure was repeated. The mice in the vehicle and LIR sub-groups were injected in the same manner with normal saline. Surgical incisions were closed by using 7-mm sterile wound clips, and an analgesic ointment was applied to the incision site. Following surgeries, mice were monitored in their recovery bins and transferred to their home bins when they were fully recovered. Throughout the procedure, the body temperature of the mice was monitored via anal probe and adjusted using a warming pad. Postoperative care was provided for 5 days to all the mice by monitoring vital signs, weight, activity, the amount of food and water ingested, as well as ensuring proper healing of the incision site. If mice showed any signs of dehydration, IP injections of saline were given. The wound clips were removed at 10 days following the surgery.

### 4.4. Animal Treatments

STZ was prepared fresh by dissolving 3 mg/kg of it into normal saline, which was injected via i.c.v. route. Sixteen days after STZ/saline injections, the mice in LIR sub-groups were injected once a day with LIR (25 nM/kg, dissolved in normal saline) intraperitoneally (IP) [18,24,44]. At the same time, the mice in vehicle and STZ groups were injected with saline IP for 30 days. The treatment protocol was adapted with slight modifications from previous studies [18,59]. Examiners were blind to the identity of animals throughout the study.

### 4.5. Behavioral Testing

All animals were subjected to behavioral testing one week prior to euthanasia. LIR injections were continued throughout the behavioral testing. Researchers conducting the behavioral tests were blinded to the group identity of the animals.

#### 4.5.1. Open Field Task

Spontaneous motor activity and exploratory behaviors of the mice were assessed in the open-field task following the protocols of Matchynski and colleagues (2013) [60,61]. In the open-field task, mice were placed in a clear 26- × 46- × 30- cm polyethylene bins for 30 min during the dark cycle. Using an automated, infrared array activity monitor (Kinder Scientific, Poway, CA, USA), spontaneous motor activity, along with time spent in the center and peripheral areas of the bin was assessed [61]. These parameters were quantified by computerized software (Kinder Scientific, Poway, CA, USA), via breaks in the infrared light beams of the apparatus. This task also served to habituate the mice for the novel-object-recognition tests, which were subsequently conducted in the same apparatus.

#### 4.5.2. Novel Object Recognition

The object-recognition-task was conducted 24 h after open-field testing, following procedures as described previously [61]. The animals were placed into 26- × 46- × 30- cm polyethylene bins for a 10-min acclimation trial. Two small objects identical in shape and color, were placed about 3 cm from the wall in the opposite corners of the 26-cm side of the bin. These acclimation sessions allowed the mice to explore the objects freely and were followed by a 5-min rest period in their home bins of the mice. Following the 5-min rest period, the mouse was placed into the experimental bin, in which one of the previously explored objects was randomly replaced with a novel object, one of the same size but a different shape. This final session lasted for 5 min, and the number of entries of the mouse into the familiar- and novel- object zones was measured.

#### 4.5.3. Passive Avoidance (PA)

The PA task measures learning and memory of the mice. The procedures used for this task were adapted from previous protocols [62,63]. This task consisted of a step-through box (Kinder Scientific, Poway, CA, USA) with light and dark chambers, a sliding door separating the chambers, stainless-steel grids on to the floors of each chamber (with only the grid in the dark chamber being electrified). Day 1 consisted of a 10-min acclimation period, in which animals were habituated to the testing apparatus by freely exploring the light and dark chambers of the apparatus. Upon entrance into the dark chamber, the sliding door was closed, and exploratory behavior was recorded. The animals remained in the chamber for the rest of the trial. On day 2, the mice were placed in the light chamber and could freely explore the chambers for 10 min with the sliding door open. Upon entrance into the dark chamber, the sliding door was closed, and following a 2-s delay, the mice were subjected to a 0.5 mA shock, lasting 2 s. The mice were left in the cage for 10 s following the shock, at which time the trial was completed and the animals were placed back in their home cages. This procedure was repeated on days 3, 4 and 5. The latency to enter the dark chamber was the primary dependent variable.

### 4.6. Tissue Processing

For histological studies, mice were deeply anesthetized with an overdose of sodium pentobarbital (0.1 mL/kg, IP), and transcardially perfused with 0.1 M cold phosphate buffer saline (PBS), followed by 4% paraformaldehyde (PFA) diluted in 0.1 M PBS at pH 7.4 to fix the brains. The brains were then removed, stored in 4% PFA until further processing. For fresh tissue samples, mice were sacrificed by cervical dislocation. The brains were then extracted, and cortical and hippocampal tissue was dissected, flash frozen in liquid nitrogen, and stored at −80 °C, until further use.

### 4.7. Immunohistochemistry (IHC)

#### 4.7.1. IHC Procedures

For immunohistochemical analyses, glial fibrillary acidic protein-GFAP and ionized calcium binding adapter molecule 1 (Iba-1) antibodies were used for labelling activated astrocytes, activated microglia respectively, whereas for labelling Aβ, 6E10 antibody was used. For the GFAP immunofluorescence protocol [64], 30-µm thick coronal sections were obtained using a cryostat (Leica; Wetzlar, Germany). Sections were rinsed three times with 1 mM PBS for 10 min each. The sections were then blocked with 10% normal goat serum (NGS) for 1 h at room temperature and were transferred to wells containing GFAP antibody (1:500) in NGS and PBS (1 mM, Ph 7.4) along with 10% NGS and 0.1% Triton X-100 and incubated overnight at 4 °C with continuous agitation. The following day, the brain sections were rinsed three times with tris-buffer saline with 0.1% Tween-20 (TBST). After three washes, the sections were incubated with anti-rabbit secondary antibody (1:1000), conjugated with Alexa-fluor-594 (Molecular Probes, OR) for 60 min at room temperature in dark, with gentle agitation. Sections were washed thoroughly with TBST and distilled water, mounted, air dried, and cover-slipped using anti-fading fluoromount aqueous mounting media, and visualized using a fluorescence microscope (Leica; Wetzlar, Germany) with appropriate excitation and emission filters

For FJB staining, 30-µm thick coronal sections were mounted on charged microscopic slides. The sections were rehydrated with distilled water and then treated with 0.006% potassium permanganate for 45 min with gentle agitation. The slides were then washed three times with distilled water, 5 min for each wash, and then treated with 0.001% FJB working solution in 0.1% acetic acid in deionized water for 30 min, with agitation. Following this, the slides were washed, air dried and cover slipped with anti-fading fluoromount aqueous mounting medium and visualized using fluorescence microscope (Leica; Wetzlar, Germany) with appropriate excitation and emission filters.

For the immunoperoxidase protocol [65], for Iba-1 and 6E10, 30-µm thick coronal sections were washed three times with 0.1M PBS and then treated with 3% hydrogen peroxide solution, dissolved in 0.1 M PBS for 30 min at room temperature to inhibit endogenous peroxidase. The sections were then rinsed with PBS and blocked with 10% NGS mixed with PBS-triton for 1 h at room temperature on a bench-top shaker to provide gentle agitation. The tissue was then incubated with Iba-1 (1:3500) at room temperature for 4 h and then kept overnight at 4 °C in the 6E10 antibodies (1:500). The following day, sections were washed three times with TBST and incubated with anti-rabbit (for Iba-1) and anti-mouse (for 6E10) biotinylated secondary antibodies for 1 h at room temperature. After three washes with TBST, the sections were incubated with peroxidase substrate solution supplied with ABC kit, and the signal was developed using diaminobenzidine (DAB), until the desired staining intensity was noted. The tissue was then washed with distilled water, cleared, mounted on to positive-charged slides, air dried and cover slipped using DePex mounting media. The sections were visualized using a compound light microscope (Olympus; Tokyo, Japan).

#### 4.7.2. Quantification of IHC Images

Quantification of GFAP-, Iba-1-, and FJB- IR cells, along with markers for Aβ were taken from four randomly selected areas of 6–9 tissue sections from the cortex and the CA1 and CA3 regions of the hippocampus. Images were captured using 20× objective (total magnification 200 times) on the Leica (for GFAP and FJB) and Olympus (for Iba-1 and Aβ. These images were analyzed using Image-J software (http://imagej.nih.gov/ij). The total area of each image was measured, and the number of GFAP-, Iba-1-, and FJB- IR cells, and the number of Aβ plaques were counted by three researchers who were blinded to the group identity of the samples. Only clearly visible large red fluorescent signals for GFAP and green fluorescent signals for FJB or DAB-labelled, brown-colored cells were counted as GFAP-IR, FJB-positive, Iba-1-IR cells respectively. The number of IR cells were expressed per 100 µm^2^ area. Samples from 3 different mice in each group were used for each parameter that was tested.

### 4.8. Western Blot

The tissue collected for Western blot analyses was processed, as described previously [57]. Tissue was homogenized, using homogenizer and pre-chilled radioimmunoprecipitation buffer (RIPA, containing 10 mM Tris, pH 8.0, 140 mM of Nacl, 30 mM of NaF, 2 mM sodium orthovanadate, 1 mM of protease cocktail inhibitors). The homogenate was centrifuged at 17,000× *g* at 4 °C for 20 min. Then, 20 µL of the supernatant was aliquoted into PCR tubes and stored at −80 °C until further use. Protein concentrations for each sample were measured using the Pierce BCA protein assay (Appendix A). Samples were mixed with equal amounts of 2× SDS-sample buffer and boiled for 3 min at 100 °C. Then, 150 µg of protein from each sample was loaded on a 4–20% gradient (for Aβ protein, IDE, pIR) and by 10% (for pAKT, and pGSK3β) tris-glycine gel and separated by SDS-polyacrylamide gel electrophoresis and then transferred on to a polyvinylidene difluoride (PVDF) membrane (Millipore-Sigma, St. Louis, MO, USA). The membrane was blocked by 5% skim milk powder dissolved in TBST and gently agitated using a shaker at room temperature for 1 h. The membranes were then incubated overnight at 4 ^0^C using the following primary antibodies (1:1000): Mouse anti-Aβ (6E10); mouse anti-IDE; rabbit anti-phospho-AKT; mouse anti-AKT; rabbit anti-GSK3β; rabbit anti-phospho-GSK3β; and rabbit anti-phospho-IR (Appendix A). The next day, blots were washed with TBST for 3 times. Membranes were then incubated with HRP-conjugated goat anti-rabbit IgG or goat anti-mouse IgG secondary antibodies (1:20,000; Santa Cruz Biotechnology, Dallas, TX, USA) mixed in 1.5% skim milk powder dissolved in TBST for 1 h at room temperature. GAPDH was used as a loading control for all the Western blot studies. Following this, the membranes were washed thoroughly three times with TBST. The blots were then developed with Western Chemiluminescent HRP-substrate by Pierce ECL Plus (Thermo-Fisher Scientific, Waltham, MA, USA). The optical densities of the bands were measured using Image-J analysis software (http://imagej.nih.gov/ij).

### 4.9. Statistical Analyses

All statistical analyses were performed using SPSS v24. All the data were expressed as mean ± SEM. The latency to escape in passive avoidance task was analyzed using repeated measures analysis of variance (ANOVA), while one-way ANOVA was used to analyze open-field and NO-recognition data. IHC and Western-blot data were analyzed using one-way analysis of variance (ANOVA) with Tukey HSD (honestly significant difference) post hoc tests conducted, when appropriate. The alpha value is set at *p* < 0.05 for all analyses.

## 5. Conclusions

Although our understanding of the link between AD and T2DM has increased considerably in recent years, there is still an unmet requirement for an effective therapeutic approach. Our study provides the first comparison of the prodromal pathological abnormalities between SAD and the 5xFAD mice. Our results indicate that neuroinflammation, and the disrupted insulin-signaling pathway was alleviated by treatment with LIR to a greater extent in prodromal STZ/sporadic AD mice than in prodromal 5xFAD mice, while Aβ pathology was alleviated by treatment with LIR to a greater extent in 5xFAD mice than in prodromal SAD. Our results support the hypothesis that LIR is a promising therapeutic agent, with significant anti-inflammatory and anti-amyloid properties, which could reduce and/or delay the onset of symptoms in prodromal stages of AD, especially in the sporadic form of the disease.

## Figures and Tables

**Figure 1 ijms-22-00860-f001:**
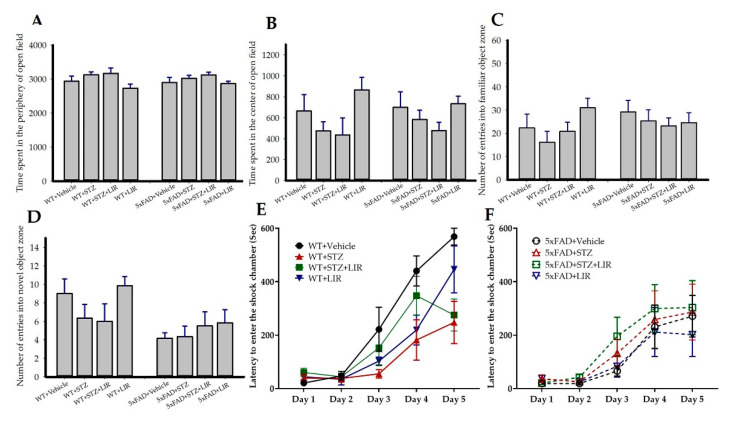
Behavioral analyses in sporadic AD model and 5xFAD mice after treatment with liraglutide: After 30 days of LIR treatment, the mice in both SAD and 5xFAD mouse models were subjected to a series of behavioral tests. Open-field (OF) task was used to assess spontaneous motor activities, exploratory behaviors were assessed through novel object recognition (NOR) task, and the differences in learning and memory were assessed through passive avoidance (PA) task. There were no significant between-group differences in the total time spent in the periphery (**A**) and in the center (**B**) of the open-field task over 60 min and in the number of entries into the familiar-object (FO; (**C**)) and novel-object (NO; (**D)**) zone in the NOR task. To test the differences in learning and memory on PA task, the ability of the mice to avoid an aversive stimulus, as measured in latency to enter the shock chamber, was analyzed by repeated-measures ANOVA. There were no significant between-group differences in the latency to avoid the shock in all trials between SAD and 5xFAD mouse models ((**E**,**F**), respectively). Data from *n* = 6 animals from each group were expressed as mean ± SEM.

**Figure 2 ijms-22-00860-f002:**
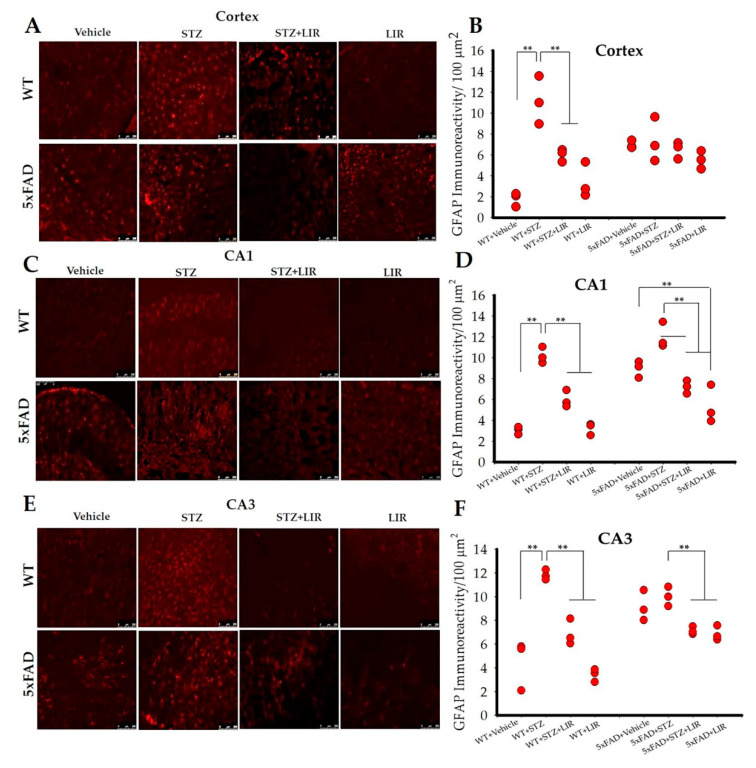
Glial fibrillary acidic protein (GFAP) immunoreactivity in the cortical, and hippocampal CA1 and CA3 regions of WT and 5xFAD mice. Following 30 days of LIR injections, WT and 5xFAD mice (*n* = 3 from each group) were sacrificed and perfused transcardially with 4% paraformaldehyde. The brains were removed and sectioned at 30 µm on a cryostat and the sections were immunolabelled with GFAP antibody. (**A**,**C**,**E**): Photomicrographs showing immunoreactive (IR) astrocytes (red) in the cortical, CA1 and CA3 regions of hippocampus in the wild type and 5xFAD mice. (**B**,**D**,**F**): Histograms representing GFAP-IR/100 µm^2^ in all the groups of the cortical, CA1 and CA3 hippocampal regions of WT and 5xFAD mice. Data from three different mice in each group were expressed as mean ± SEM; Scale bar is 250 µm for all images. ** *p* < 0.01.

**Figure 3 ijms-22-00860-f003:**
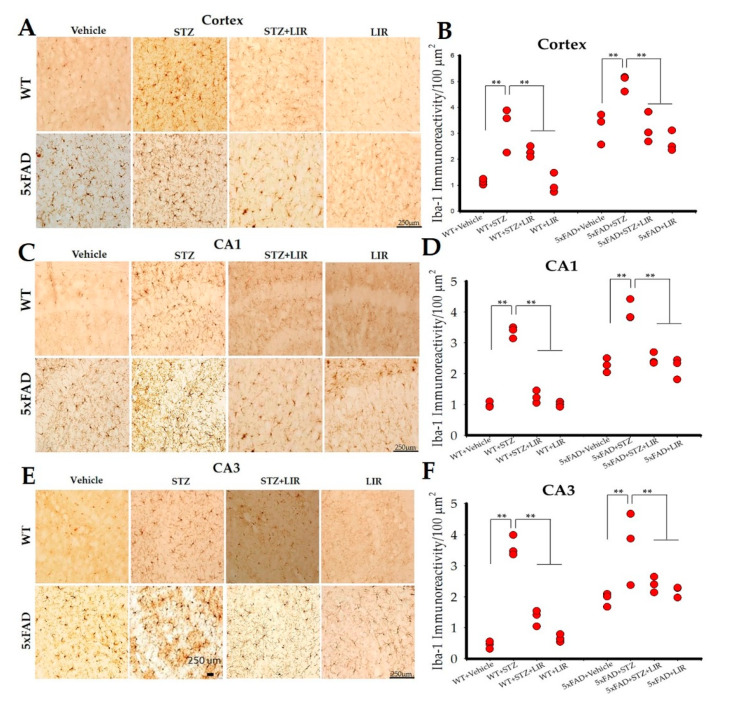
Microglial activation in the cortical, and hippocampal CA1 and CA3 regions of WT and 5xFAD mice. Following 30 days of LIR injections, WT and 5xFAD mice (*n* = 3 from each group) were sacrificed and perfused transcardially with 4% paraformaldehyde. The brains were sectioned at 30 µm on a cryostat and the sections were immunolabelled with Iba-1 antibody. (**A**,**C**,**E**): Photomicrographs showing immunoreactive (IR) microglia (brown) in the cortical, CA1 and CA3 regions of hippocampus in the WT and 5xFAD mice. (**B**,**D**,**F**): Histograms representing Iba-1-IR/100 µm^2^ in all the groups of the cortical, CA1 and CA3 hippocampal regions of WT and 5xFAD mice. Data from three different mice in each group were expressed as mean ± SEM; Scale bar is 250 µm for all images. ** *p* < 0.01.

**Figure 4 ijms-22-00860-f004:**
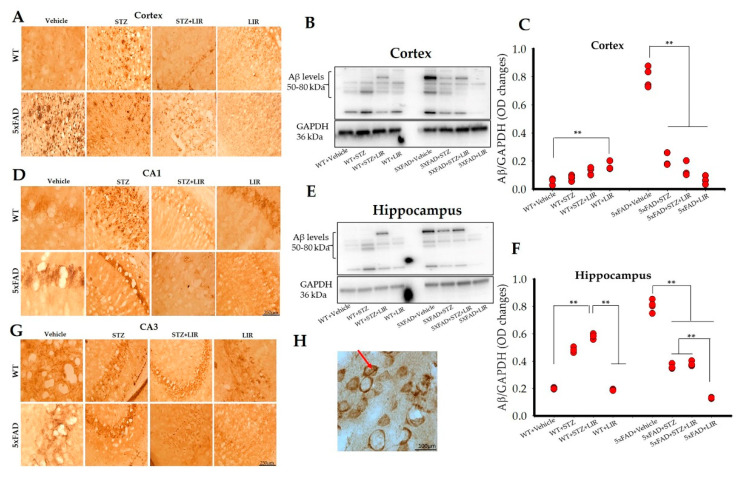
Aβ protein levels in the cortical and hippocampal regions of the WT and 5xFAD mice. Following LIR treatment, WT and 5xFAD mice (*n* = 3 from each group) were sacrificed and perfused transcardially with 4% paraformaldehyde and sectioned at 30 µm on a cryostat and the sections were immunolabelled with 6E10 antibody. (**A**,**D**,**G**): Representative 6E10 immunoreactivity for Aβ levels in the cortical (**A**), CA1 (**D**) and CA3 (**G**) areas of WT and 5xFAD mice. Immunopositive Aβ (brown) were recognized and stained with 6E10 antibody. (**H**): Representative 100× image showing intraneuronal inclusions (red arrow) in CA1 region of 5xFAD mouse model. Following LIR treatment, WT and 5xFAD mice (*n* = 3 from each group) were sacrificed via cervical dislocation, fresh brains were collected, tissue was homogenized after which Western blot analysis was performed. (**B**): Representative Western blot bands showing Aβ levels in the cortex of both WT and 5xFAD mice. (**C**): Histogram showing densitometric quantification (mean ± SEM) done on Aβ (from B) normalized with GAPDH. (**E**): Western blot bands showing Aβ levels in the whole hippocampus of WT and 5xFAD mice. (**F**): Histogram showing densitometric quantification (mean ± SEM) done on Aβ protein (from E) normalized with GAPDH. Scale bar is 250 µm for all images in A, D and G. Scale bar for 100× image is 100 µm. ** *p* < 0.01.

**Figure 5 ijms-22-00860-f005:**
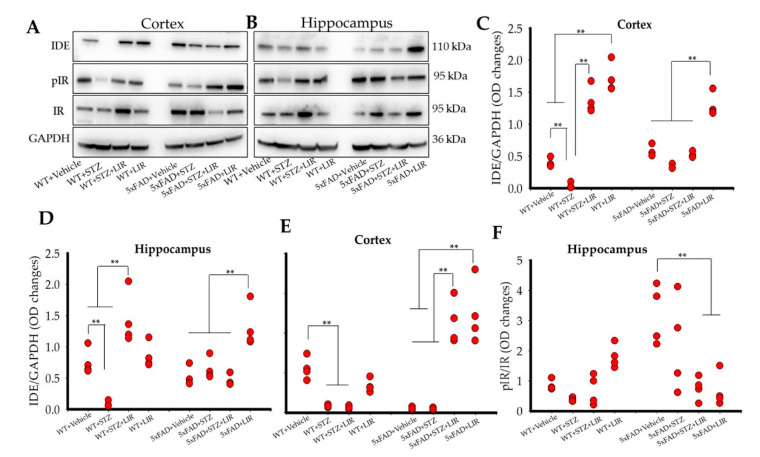
Insulin-Degrading Enzyme and Phosphorylated Insulin Receptor protein levels in the cortical and hippocampal regions of the WT and 5xFAD mice. Following LIR treatment, WT and 5xFAD mice (*n* = 3 from each group) were sacrificed via cervical dislocation and their brains were immediately collected and homogenized for Western blot analysis. (**A**,**B**): Representative Western blot bands for IDE, pIR, and IR in both cortical and hippocampal regions of WT and 5xFAD mice. (**C**,**D**): Histograms showing densitometric quantification of IDE, normalized with GAPDH, in the cortex and hippocampal regions of the brain, respectively. (**E**,**F**): Histograms showing densitometric quantification of pIR normalized with GAPDH and IR in the cortex and hippocampal regions of the brain, respectively. ** *p* < 0.01.

**Figure 6 ijms-22-00860-f006:**
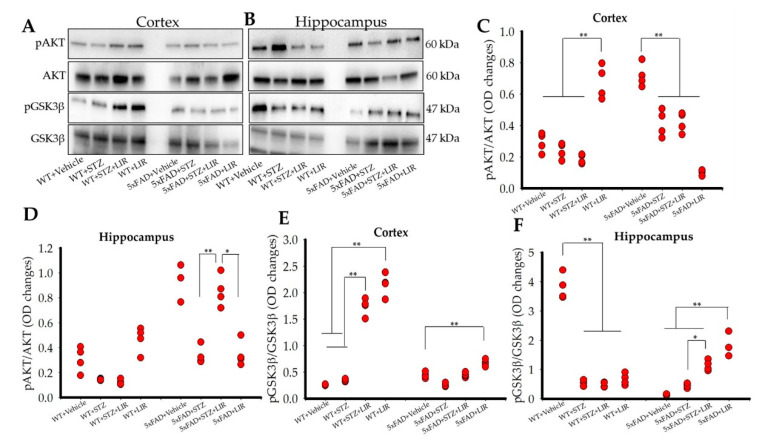
pAKT and pGSK3β protein levels in the cortical and hippocampal regions of both WT and 5xFAD mice. Following LIR treatment, WT and 5xFAD mice (*n* = 3 from each group) were sacrificed via cervical dislocation and their fresh brains were immediately collected and homogenized for Western blot analysis. (**A**,**B**): Representative Western blot bands for pAKT and pGSK3β in both cortical and hippocampal regions of WT and 5xFAD mice. (**C**,**D**): Histograms showing densitometric quantification of pAKT, normalized with GAPDH and AKT in cortex and hippocampal regions of the brain respectively. (**E**,**F**): Histograms showing densitometric quantification of pGSK3β, normalized with GAPDH and GSK3β in the cortical and hippocampal regions of WT and 5xFAD mice brain tissue. * *p* < 0.05. ** *p* < 0.01.

**Figure 7 ijms-22-00860-f007:**
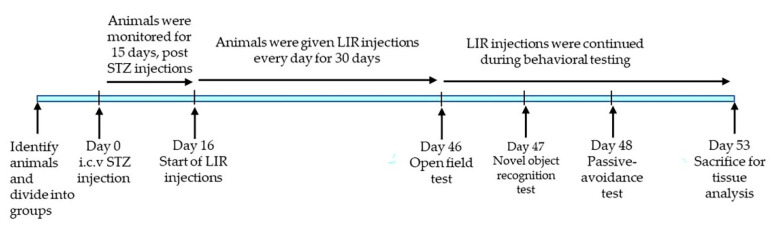
Schematic representation of the experimental design and treatment timeline of the study.

## Data Availability

Data is contained within the article or Appendix A.

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
