# Peer review of "Liraglutide Has Anti-Inflammatory and Anti-Amyloid Properties in Streptozotocin-Induced and 5xFAD Mouse Models of Alzheimer’s Disease"

_ijms, 2021, doi:10.3390/ijms22020860_

Round 1

Reviewer 1 Report

The manuscript by Paladugu and colleagues investigate the anti-inflammatory and anti-amyloid properties of liraglutide, an anti-diabetic agent in a sporadic and mixed AD animal model of the disease. The authors use STZ administration into the lateral ventricles of 3-month-old wild-type mice in order to mimic sporadic AD and commence treating these mice (16 days following STZ) with LIR for 30 days. A similar procedure was also performed on 3-month-old 5XFAD mice. This component (5XFAD-STZ) of the study is confusing. Why were 5XFAD mice, a genetic model of the disease, treated with STZ? This arm of the study does not make sense and has confounded the results. As it currently stands, this study does not warrant publication in IJMS. Major: - What was the justification for administering STZ to 5XFAD mice? - What was the dose of STZ that was administered per animal? - How long did mice in the STZ group receive LIR? - When were the behavioral tests performed? Why were LIR injections administered during the behavioral testing? - Please provide a clear timeline: indicate when animals received STZ and LIR and when behavioral tests were performed. - There are contradictory comments made in the document: Ln 373 – “our behavioral analyses was completed before the mice reached 4.5 months of age” Ln 439 – “Since the mice were only 4.5 months of age at the time of analyses” Calculations suggest that mice would have been 4.5 months at the time of behavioral testing. As the authors state, studies (including those in the reviewers laboratory) have shown that cognitive deficits in 5xFAD mice are present at 4 months of age and as such, the present findings are somewhat surprising. - Fig 1B – results demonstrate that LIR administration to WT mice result in a statistically significant increase in activated astrocytes – this must be explained in the manuscript. - Fig 1A, C, E – these images are of very poor quality and higher quality images must be provided. - Fig 3 – Can the authors please clarify what has been quantified in 3C and 3F? Is it "Abeta levels 50-80kDa" or the lower species at the bottom of the membrane? - Fig 3B and 4 - These results rely on western blotting experiments. The authors have used GAPDH as the ‘housekeeping’ gene and quantified the expression level of proteins using GAPDH as the loading control. The basic principal of using a loading control is to examine whether equal amounts of protein have been loaded across all samples. In order to achieve this, the immunoreactive signal of the control protein must be within the logarithmic phase of expression where loading differences can be visible to the naked eye. The GAPDH signals in blots shown in this manuscript other than 3E is supersaturated and it is therefore not clear how the authors were able to detect any changes in protein expression. All proteins under investigation must be normalized to GAPDH. If the GAPDH signal is supersaturated, this task is not possible. The authors need to re-run these western blots and expose membranes probed for GAPDH for significantly shorter periods of time (similar to 3E) before any claims can be made. - From this reviewers personal experience using antibody 6E10, Abeta is not detected by WB in WT mice. As such, it is somewhat surprising to see such high levels of Abeta in WT mouse brain. Can the authors please confirm that this is not due to an overflow from adjoining lanes or from contaminated sample preparation considering that samples were prepared via homogenization? A good test for this would be to homogenize RIPA buffer (without tissue samples) using the homogenizer and testing via ELISA or dot blots. - Fig 3C – The authors must comment on why there is such a large, statistically significant difference in Abeta levels between 5XFAD-Vehicle and 5XFAD-STZ in both the cortex and hippocampus when one would assume that the administration of STZ would exacerbate Abeta levels in the brain. These results appear to suggest that STZ suppresses Abeta accumulation in the Tg brain. If the latter, then are the results using LIR valid? - Additionally, administration of LIR to WT-STZ mice appear to have no effect – please comment in discussion. - Fig 4A – Can the authors please confirm that treating WT mice with STZ resulted in the complete obliteration of IDE expression in all 6 animals? - In the discussion the authors state they “found that following STZ injections, there was increased accumulation of intraneuronal Aβ inclusions along with a few extracellular Aβ plaque deposits in both SAD and 5xFAD mouse models” and that “the extent of extracellular plaque deposition following STZ injections was more pronounced in SAD mice when compared to 5xFAD model”. This is however not correct. The authors did not count plaque numbers and WB results do not support these comments. - Ln 384 – “LIR treatment reduced the number of plaques in both SAD and 5xFAD mice”. This statement is not substantiated as plaque counting was not performed. - Ln 437 – The authors state they “observed a reduced expression of pAKT in both the cortical and hippocampal regions in both AD models following STZ injections, but treatments with LIR elevated pAKT levels in the SAD mice but not 5xFAD mice”. This is an incorrect/false statement. LIR resulted in a further reduction in pAKT levels (Fig 5C) in WT-STZ mice and in the 5XFAD mice as well. The only increase was observed in WT-LIR mice. - The discussion is very confusing as the authors have attempted to use existing literature to justify most of their results. There appears to be significant confusion as to how and why the results have played out the way they have. The administration of STZ to 5XFAD mice have confounded the confusion and the authors are advised to remove this arm of the study. Minor: Table 2 should be moved to the supplementary information section of the manuscript.

Author Response

Response to Reviewer 1 comments

We thank the reviewer for their thorough revision of the manuscript and for the critical comments which helped us improve the manuscript.

Point 1: What was the justification for administering STZ to 5XFAD mice?

Response 1: One of the aims of the study was to create a SAD mouse model by injecting WT mice with STZ and to compare the pathological changes of SAD model with AD transgenic models, such as 5xFAD. Previous data suggested that the early stages of 5xFAD mice (3-4 months) does not produce much extracellular amyloid plaque, therefore, we were interested to produce more extracellular Aβ plaque deposition in 5xFAD mice at early stage by injecting these mice with STZ and to investigate and compare the anti-amyloid properties of LIR in both SAD and 5xFAD mice models. That is why we injected STZ in 5xFAD mice.

Point 2: What was the dose of STZ that was administered per animal?

Response 2: STZ was administered at a dose of 3 mg/kg body weight as reported by Xiong and colleagues (please see Ref 45). This was mentioned in the methods section of the revised manuscript, please see page 15, line 563.

Point 3: How long did mice in the STZ group receive LIR?

Response 3: The mice in the STZ group received 30 days of LIR treatment. Additionally, the mice received LIR injections during behavior testing. This was represented as a flow chart (Figure 7) in the methods section of the revised manuscript, please see page 15, lines 549-550.

Point 4: When were the behavioral tests performed?

Response 4: Behavior tests were started at noon every day, after the mice received 30 days of LIR injections. The testing continued for 7 days. This information is documented in Figure 7 in the methods section of the revised manuscript. Please see page 15, line 549-550.

Point 5: Why were LIR injections administered during the behavioral testing?

Response 5: The half-life of LIR is 13 hrs, and when injected once a day, LIR maintains its therapeutic effect steadily. It takes 3 days for LIR to be eliminated completely from the system after the last dose. Therefore, if LIR injections were stopped after 30 days and not continued during behavioral testing, the effects of the drug would be weaned off totally and we were interested to measure the behavioral, biochemical and physiological outcomes in presence of LIR. Therefore, to maintain the steady state of LIR concentrations in the body, LIR injections were continued during the behavioral testing.

Point 6: Please provide a clear timeline: indicate when animals received STZ and LIR and when behavioral tests were performed.

Response 6: We have added a flowchart describing the experimental timeline. Please see in the methods section of the revised manuscript, page 15, lines 549-550.

Point 7: There are contradictory comments made in the document: Ln 373 – “our behavioral analyses was completed before the mice reached 4.5 months of age” Ln 439 – “Since the mice were only 4.5 months of age at the time of analyses” Calculations suggest that mice would have been 4.5 months at the time of behavioral testing. As the authors state, studies (including those in the reviewer’s laboratory) have shown that cognitive deficits in 5xFAD mice are present at 4 months of age and as such, the present findings are somewhat surprising.

Response 7: When we started our experiment, the mice were 3 months old. By the end of LIR injections, the mice were 4.5 months old and at the end of behavioral testing they were 4.7 months. We have clarified this confusion in the revised manuscript by adding these sentences in the discussion section of revised manuscript as “our behavioral analyses were completed before the mice reached 4.5 months of age” was changed to “our behavioral analyses were completed at 4.7 months of age”. Please see page 12, lines 411-412.

Different testing parameters between studies may be the reason for the discrepant findings between studies.  The most salient difference may be that the repeated testing conducted in this study, especially PA task, at this early age may have protected against deficits that would have otherwise emerged with time. This explanation was also mentioned in the discussion part of the revised manuscript. Please see page 12, lines 413-416.

Point 8: Fig 1B – results demonstrate that LIR administration to WT mice results in a statistically significant increase in activated astrocytes – this must be explained in the manuscript.

Response 8: We sincerely apologize to the reviewer for a mistake from our end. When we were changing the graphs to new ones, we noticed there was no significant difference in the number of astrocytes between WT+Vehicle and WT+LIR group. We have rectified the mistake. Please see Fig 2 B of the revised manuscript, page 5, line 168.

Point 9: Fig 1A, C, E – these images are of very poor quality and higher quality images must be provided.

Response 9: We have added high resolution images of Fig 1A, C and E in the revised manuscript. Please see revised Fig 2A, C, and E. Please see page 5, line 168.

Point 10: Fig 3 – Can the authors please clarify what has been quantified in 3C and 3F? Is it "Abeta levels 50-80kDa" or the lower species at the bottom of the membrane?

Response 10: We have quantified the Aβ levels at 50-80 kDa. The species at the bottom of the membrane have a molecular weight of around 15-20 kDa, they are the tetramers and the pentamers.

Point 11: Fig 3B and 4 - These results rely on western blotting experiments. The authors have used GAPDH as the ‘housekeeping’ gene and quantified the expression level of proteins using GAPDH as the loading control. The basic principle of using a loading control is to examine whether equal amounts of protein have been loaded across all samples. In order to achieve this, the immunoreactive signal of the control protein must be within the logarithmic phase of expression where loading differences can be visible to the naked eye. The GAPDH signals in blots shown in this manuscript other than 3E is supersaturated and it is therefore not clear how the authors were able to detect any changes in protein expression. All proteins under investigation must be normalized to GAPDH. If the GAPDH signal is supersaturated, this task is not possible. The authors need to re-run these western blots and expose membranes probed for GAPDH for significantly shorter periods of time (similar to 3E) before any claims can be made.

Response 11: Using auto-mode, we have developed GAPDH immunoblot in a gel documentation system (Bio-Rad) which we have standardized in our laboratory with many publications. We have taken care to make sure the signal for GAPDH was not supersaturated. Also, we normalized all the proteins (including phosphorylated forms) under investigation with GAPDH. Fig 5 & 6, in page 10 and 11 of the revised manuscript showed representative images of phosphorylated proteins and their respective total forms.

Point 12: From this reviewer's personal experience using antibody 6E10, Abeta is not detected by WB in WT mice. As such, it is somewhat surprising to see such high levels of Abeta in WT mouse brains. Can the authors please confirm that this is not due to an overflow from adjoining lanes or from contaminated sample preparation considering that samples were prepared via homogenization? A good test for this would be to homogenize RIPA buffers (without tissue samples) using the homogenizer and testing via ELISA or dot blots.

Response 12: Measures were taken to prevent contamination of the tissue, right from dissection of the tissue, to sample loading into the SDS-page gel. We made sure the dissection tools were cleaned with water and ethanol, and air dried before proceeding to the next animal. The same procedure was followed during homogenization. While loading the sample into the wells, care was taken to avoid overloading the sample into the well. The maximum capacity of the wells was 40 µL, and an average sample volume we loaded into the wells was 5 µL. We thought that the detection of Aβ levels in the WT+Vehicle mice might be due to the presence of soluble Aβ species, which can be detected by 6E10 antibodies. As such, we noticed a strong band around 15-20 kDa region (tetramer to pentamer) on the membrane and almost no band on the top part of the membrane (an area for low to high molecular weight Aβ species, Fig 4B and E).

Point 13: Fig 3C – The authors must comment on why there is such a large, statistically significant difference in Abeta levels between 5XFAD-Vehicle and 5XFAD-STZ in both the cortex and hippocampus when one would assume that the administration of STZ would exacerbate Abeta levels in the brain. These results appear to suggest that STZ suppresses Abeta accumulation in the Tg brain. If the latter, then are the results using LIR valid? - Additionally, administration of LIR to WT-STZ mice appears to have no effect – please comment in discussion.

Response 13: We assumed that STZ would increase the Aβ levels when injected in 5xFAD mice, but we did not observe it. However, when we compared the 5xFAD+Vehicle mice and 5xFAD+LIR mice, we observed a significant LIR-induced decrease in the levels of Aβ in both cortical and hippocampal tissue. We think this is a very important finding from this study which eventually validated the use of LIR. This observation warrants further detailed study in order to discern the role of STZ in transgenic mice. In SAD mice, as the reviewer pointed out that administration of LIR to WT+STZ group did not have a significant effect. This might be due to the relatively lower levels of Aβ, reducing any effect size of LIR on Aβ levels. However, we assume that as the animal ages and following long term administration of LIR, there might be a significant treatment effect. When we observe the Western blot bands, the lane with WT+STZ+LIR has a strong band on the top part of the blot in both cortical and hippocampal tissue. Since we analyzed all the bands together within the range of 50-80 kDa for measuring the levels of Aβ, the strong band might have contributed to the observed results in these groups. We thought these changes can be reversed with long term administration of LIR, which needs further investigation. These lines are added in the “discussion” part of the revised manuscript. Please see page 12, lines 426-435.

Point 14: Fig 4A – Can the authors please confirm that treating WT mice with STZ resulted in the complete obliteration of IDE expression in all 6 animals?

Response 14: For all our Western blot experiments, we used 3 mice per group. Also, three independent experiments were performed, and we observed similar findings with regards to the expression of IDE in WT+STZ groups in all the 3 animals as shown in the representative blots (Fig 5A).

Point 15: In the discussion the authors state they “found that following STZ injections, there was increased accumulation of intraneuronal Aβ inclusions along with a few extracellular Aβ plaque deposits in both SAD and 5xFAD mouse models” and that “the extent of extracellular plaque deposition following STZ injections was more pronounced in SAD mice when compared to 5xFAD model”. This is however not correct. The authors did not count plaque numbers and WB results do not support these comments.

Response 15: We agree with the reviewer statement. We rephrased these sentences in the revised manuscript. Please see page 12, lines 421-437.

Point 16: Ln 384 – “LIR treatment reduced the number of plaques in both SAD and 5xFAD mice”. This statement is not substantiated as plaque counting was not performed.

Response 16: We agree with the reviewer’s point. We clarified this part of discussion in the revised manuscript, please see page 12, line 421-437.

Point 17: Ln 437 – The authors state they “observed a reduced expression of pAKT in both the cortical and hippocampal regions in both AD models following STZ injections, but treatments with LIR elevated pAKT levels in the SAD mice but not 5xFAD mice”. This is an incorrect/false statement. LIR resulted in a further reduction in pAKT levels (Fig 5C) in WT-STZ mice and in the 5XFAD mice as well. The only increase was observed in WT-LIR mice.

Response 17: We agree with the reviewer about the above-mentioned contradictory statement. We re-wrote the sentence by adding these sentences in the revised manuscript as “ We also observed a reduced expression of pAKT in both the cortical and hippocampal regions in both AD models following STZ injections, but treatments with LIR did not significantly elevate the pAKT levels in the SAD mice (except in the WT+LIR group)  and in 5xFAD mice; LIR, in fact, reduced the pAKT levels in 5xFAD+LIR group when compared to 5xFAD+Vehicle group in both cortical and hippocampal regions of the brain. We noticed a trend towards LIR causing a further reduction of cortical pAKT levels in the WT+STZ+LIR mice when compared to WT+STZ group but the changes were not significant”. Please see page 13, lines 494-499 and page 14, lines 500-501 in the revised manuscript.

Point 18: The discussion is very confusing as the authors have attempted to use existing literature to justify most of their results. There appears to be significant confusion as to how and why the results have played out the way they have. The administration of STZ to 5XFAD mice has confounded the confusion and the authors are advised to remove this arm of the study.

Response 18: We have included changes in the discussion part to the revised manuscript to make it clear. In our study, the results from 5xFAD mice, especially with respect to pAKT and pIR are difficult to predict in these relatively young mice. Please see page 14, lines 501-506 in the discussion part of the revised manuscript. As the reviewer suggested, if the 5xFAD+STZ and 5xFAD+STZ+LIR arm is removed from the study, the comparison would be between 5xFAD+Vehicle and 5xFAD+LIR groups. When just these two groups were compared, we noticed a significant reduction in the Aβ levels in the 5xFAD+LIR group when compared to the 5xFAD+Vehicle group in both cortical and hippocampal regions of the brain (page 12, lines 427 and 428). We also noticed, a significant increase in the expression of cortical pIR levels in the 5xFAD+LIR group when compared to 5xFAD+Vehicle group (page 12, 488-490). However, the cortical and hippocampal pAKT levels were not increased in the 5xFAD+LIR group when compared to 5xFAD+Vehicle group. Since our Western blot images have 5xFAD+STZ and 5xFAD+STZ+LIR groups in the middle two lanes it would be difficult to represent the other groups efficiently. Therefore, we included the discussion of all the four groups of 5xFAD mice, but conclusions were made from the 5xFAD+Vehicle and 5xFAD+LIR groups only. Please see page

Point 19: Table 2 should be moved to the supplementary information section of the manuscript.

Response 19: We deleted table 2 from the revised manuscript and added to the supplementary section of the revised manuscript.

Reviewer 2 Report

Comments to Authors:

Title: Liraglutide has Anti-Inflammatory and Anti-Amyloid Properties in Streptozotocin-Induced and 5xFAD Mouse models of Alzheimer’s Disease.

Authors: Paladugu,L.; Gharaibeh, A.; Kolli, N.; Learman, C.; Hall, Tc.; Li, L.; Rossignal, J.; Maiti, P.; Dunbar, GL.

Overview:

The manuscript presents an in vivo study of the drug liraglutide as a potential AD therapeutic. The study involved 5xFAD transgenic and SAD mouse models, with streptozotocin used to induce insulin resistance and model SAD in wild-type mice. Assessments of AD pathophysiology such as Amyloid-beta deposition along with behavioral assessments, though tests were conducted prior to the typical onset of cognitive decline in mouse models of AD. No significant differences were found in cognition/behavior, though Aβ accumulation and neuroinflammation that typically precedes cognitive decline were noted in the FAD and SAD mice, were ameliorated by liraglutide. Overall the work is well written and well explained throughout, and serves as a solid base for further investigation into liraglutide as a therapeutic agent in prodromal stage AD.

Minor concerns:

Results:

  1. Sample size could be added to table 1 (I.e., add “n = 6” next to each group).
  2. The font on some of the figures, specifically the graphs, is too small and low-resolution to be read.

Discussion:

  1. Line 394: “Increase in neuroinflammation” should be changed to “An increase in neuroinflammation.

Materials and methods:

  1. Line 625: “Protein concentrations for each sample was measured…” should be changed to “Protein concentrations for each sample were measured…”

Author Response

Response to Reviewer 2 comments

We thank the reviewer for a thoughtful review of our work and for the kind input and comments. We have revised the manuscript and incorporated the suggested changes.

Point 1: Sample size could be added to table 1 (I.e., add “n = 6” next to each group).

Response 1: As per the reviewer’s suggestion, we have added the sample size in the figure-1 legend, in the results section of the revised manuscript, please see page 4, line 129-130.

Point 2: The font on some of the figures, specifically the graphs, is too small and low-resolution to be read.

Response 2: We have made the suggested changes of all the figures in the revised manuscript. Please see all the revised figures.

Point 3: Line 394: “Increase in neuroinflammation” should be changed to “An increase in neuroinflammation

Response 3: We included the suggested change in the revised manuscript. Please see page 13, line: 446.

Point 4: Line 625: “Protein concentrations for each sample was measured…” should be changed to “Protein concentrations for each sample were measured…”

Response 4: We included the suggested change in the revised manuscript. Please see page 18, lines 695.

Reviewer 3 Report

In this work authors tried to explore the anti-diabetic drug liraglutide (LIR) in terms of its anti-inflammatory and anti-amyloid properties (this already known) in the context of two AD models. They tried to do this in the early stages of the disease, however they could not demonstrate this clearly in the behavioral tests, neither biochemically. Regarding the effect of LIR, authors could clearly show that SVZ promoted inflammation as expected and that LIR could reduce it, in both AD models. However, the results/conclusions between WT, and two AD models was never clear, and mixed wrong conclusions, in my view, were taken. Authors claim that this early intervention with anti-diabetic drugs might prove beneficial in reversing the neuroinflammatory and amyloid pathology in prodromal AD. So they should have done some longer study to prove it, in one of the models, e.g. with behavioral experiments.

- Main findings of the study:

. The authors wanted to address the therapeutical role of liraglutide (LIR), an anti-diabetic drug, in the context of Alzheimer Disease, since diabetes mellitus is a strong risk factor for the development of Alzheimer’s disease

. To access the effects of LIR in AD authors used two AD animal models: a transgenic (5xFAD) and a sporadic model (injecting streptozotocin in lateral ventricles). 30 days LIR treatment

. Authors accessed neurodegeneration by Aβ plaque load, signaling pathways and behavioral tests.

. Authors found that LIR was able to decreased neuroinflammatory responses in both SAD and 5xFAD mice before significant cognitive changes were observed.

. Authors claim that LIR has inflammatory and anti-amyloid prophylactic therapy in the prodromal stages of AD.

- Limitations

. Authors should look/follow the ARRIVE guidelines for reporting in-vivo animal experiments. They contain important recommendations. Some points are missing such as a time-line or flow chart illustrating the study, exclusion criteria (in animal model), % of animal death, blinding (very important in data quantification and behavior), experimental unit, among others.  However, the methodological section is already nicely presented and carefully detailed.

. I could not easily find the number of independent repetitions for each experiment (!!), neither in methods, results or legends, for some of the experimental approaches (Fig 1-5)

. The organization of the conditions in the graphs is not adequate, in my view, to promote the right comparisons. For example Wt+LIR should be next to the Wt+Vehicle; Wt+SVZ with Wt+SVZ+LIR; …

. Behavioral experiments are badly presented, avoiding correct result interpretation

Major revisions:

. All the points raised in Limitations section that refer to important issues. They have to be addressed, and corrected, when possible.

. Table 1 should be presented in graphs to better data interpretation, since this also an important part of the paper. This way result interpretation is painful and a lot of information is missing! When possible representative tracings would be an advantage to result analysis/interpretation.

. Under Table 1 results also, authors mention that “No between-group differences were observed on any behavioral measures”. Since authors want to study AD in its prodromal stages, it is important to show that for example 5xFAD mice show normal behavior. It is known by 4–5 months they start to exhibit cognitive deficits. In the studied 3 months what is happening?

. The authors mention in line 215 figure 3 that “LIR treatment significantly decreased the expression of hippocampal Aβ levels in the WT+LIR group when compared to WT+STZ+LIR group”. I cannot see the results that lead to that conclusion, in the quantification of ABeta load in Fig. 3C and F. Moreover, when comparing SVZ whith SVZ+LIR condition, LIR couldn’t decrease ABeta levels.

. In Figure 4 authors claim that Liraglutide increased the levels of insulin-degrading enzyme (IDE) and phosphorylated insulin-receptor (pIR) in the cortical and hippocampal regions of WT and 5xFADmice. Fig. 4F shows that pIR is not upregulated by LIR in the hippocampus

. In Figure 5 authors claim that LIR increase the levels of pAKT and pGSK3β in the cortical and hippocampal regions of WT and 5xFAD mice. When analyzing Fig 5C and D and looking for 5xFAD and 5xFAD+LIR conditions I can only see opposite conclusion. The same for SAD mouse models of AD.

. Authors should clear the results/conclusions regarding LIR effects in physiological conditions, 5xFAD model and SAD mouse AD model. Most of the time results/conclusions are all mixed-up…as previous comments show

. Legends should be self-explained, and some data is missing regarding number of independent experiments performed, indication of the experimental unit and better description of the statistics analysis/results (for each experiment), as mentioned.

. I did not find a comparison/discussion/result integration between other similar works such as in J Pathol. 2018 May;245(1):85-100 , that describes how liraglutide reverses cognitive impairment in mice and attenuates insulin receptor and synaptic pathology in a non-human primate model of Alzheimer's disease!

Minor points:

. All the graphs (Fig. 1-5) should indicate the individual data points to understand variability within each group. This also contributes to data transparency, robustness and data interpretation.

. A final cartoon/graphical abstract/scheme would be a plus to the paper

Author Response

Response to Reviewer 3 comments

We thank the reviewer for a detailed review of our work and for the critical input which helped us improve our manuscript. We have revised the manuscript and included the suggested changes.

Point 1: Authors should look/follow the ARRIVE guidelines for reporting in-vivo animal experiments. They contain important recommendations. Some points are missing such as a time-line or flowchart illustrating the study, exclusion criteria (in animal model), % of animal death, blinding (very important in data quantification and behavior), experimental unit, among others.  However, the methodological section is already nicely presented and carefully detailed.

Response 1: We have included a timeline chart (Fig 7) showing a schematic representation of the study in the methods section of the revised manuscript, please see page 15, lines 549-550. Our experimental unit is a single mouse and all the mice survived all the surgical and treatment interventions, so analyses were performed on all the mice included in the study. This was included in the results section of the revised manuscript, page 3, lines 101-103. The researchers conducting the behavior testing were blinded to the group identity of the animals. We have mentioned the above points in the methods section of the revised manuscript, please see page 15, lines 583-584 and 587-588. 

Point 2: I could not easily find the number of independent repetitions for each experiment (!!), neither in methods, results or legends, for some of the experimental approaches (Fig 1-5)

Response 2: Every experiment was repeated three times. The same was included in the methods section of the revised manuscript as “Samples from three different mice from each group were used for each parameter that was tested”. Please see page 17, lines 686-687. Additionally, we also included a sentence in the revised manuscript in all the figure legends as “Data from three different mice in each group were expressed as mean ± SEM”.

Point 3: The organization of the conditions in the graphs is not adequate, in my view, to promote the right comparisons. For example, WT+LIR should be next to the WT+Vehicle; Wt+STZ with Wt+STZ+LIR; …

Response 3: We wanted to check the effect of STZ in WT mice, so we opted to put the WT+STZ group right next to WT+Vehicle to make an easy comparison. To be consistent between both models, we followed the same order for the 5xFAD group also. Since our western blot representative images show the WT+LIR group at the end of the image, changing the conditions in the graphs would create confusion between images and graph interpretation. We thank the reviewer for this suggestion. We will incorporate this valuable suggestion in our future studies.

Point 4: Table 1 should be presented in graphs to better data interpretation, since this is also an important part of the paper. This way the result interpretation is painful, and a lot of information is missing! When possible, representative tracings would be an advantage to result analysis/interpretation.

Response 4: We agree with the reviewer’s view about better presentation of behavior results. Therefore, we presented all the behavior data through graphical representation. Since table 1 has the same data as Fig 1, we deleted table 1, and included Figure 1 in the revised manuscript. Please see page 3, line 113 of the revised manuscript.

Point 5: Under Table 1 results also, authors mention that “No between-group differences were observed on any behavioral measures”. Since authors want to study AD in its prodromal stages, it is important to show that for example 5xFAD mice show normal behavior. It is known by 4–5 months they start to exhibit cognitive deficits. In the 3 months studied, what is happening?

Response 5: As mentioned in the manuscript, Oakley et al, reference number 27, at 2 months of age no deficits were observed, and the 5xFAD mice start exhibiting cognitive deficits around 4-5 months of age. Therefore, we did not expect to see any deficits in 3-month-old 5xFAD mice. This was included in the discussion section of the revised manuscript. Please see page 12, lines 410-411.

Point 6: The authors mention in line 215 figure 3 that “LIR treatment significantly decreased the expression of hippocampal Aβ levels in the WT+LIR group when compared to WT+STZ+LIR group”. I cannot see the results that lead to that conclusion, in the quantification of ABeta load in Fig. 3C and F. Moreover, when comparing STZ with STZ+LIR condition, LIR couldn’t decrease ABeta levels.

Response 6: We clarified these sentences in the revised manuscript by adding the following sentences “LIR treatment significantly decreased the expression of hippocampal Aβ levels in the WT+LIR group when compared to WT+STZ+LIR group. However, the Aβ levels were slightly increased in the WT+STZ+LIR group when compared to the WT+STZ group but the changes were not significant”. Please see page 7, lines 234-237 in the revised manuscript.

Point 7: In Figure 4 authors claim that Liraglutide increased the levels of insulin-degrading enzyme (IDE) and phosphorylated insulin-receptor (pIR) in the cortical and hippocampal regions of WT and 5xFADmice. Fig. 4F shows that pIR is not upregulated by LIR in the hippocampus

Response 7: We agree with the reviewer’s point. We reworded this portion by adding these sentences in the revised manuscript “Liraglutide increased the levels of insulin-degrading enzyme (IDE) in the cortical and hippocampal regions of WT and 5xFAD mice, while liraglutide increased phosphorylated insulin receptor (pIR) in the cortical but not the hippocampal regions of the WT and 5xFAD mice”. Please see page 8, lines 265-267 in the revised manuscript.

Point 8: In Figure 5 authors claim that LIR increases the levels of pAKT and pGSK3β in the cortical and hippocampal regions of WT and 5xFAD mice. When analyzing Fig 5C and D and looking for 5xFAD and 5xFAD+LIR conditions I can only see the opposite conclusion. The same for SAD mouse models of AD.

Response 8: We thank the reviewer for catching this point. We have clarified this view by adding these sentences in the revised manuscript as “Liraglutide did not increase the levels of pAKT in the cortical and the hippocampal regions of the brain in SAD and 5xFAD mice, while liraglutide increased the levels of pGSK3β in the cortical region of both SAD and 5xFAD mice and only in the hippocampal region of 5xFAD mice”. Please see page 10, lines 332-334 in the revised manuscript.

Point 9: Authors should clear the results/conclusions regarding LIR effects in physiological conditions, 5xFAD model and SAD mouse AD model. Most of the time results/conclusions are all mixed-up…as previous comments show

Response 9: As per the reviewer's suggestion, the necessary changes were made to the results and conclusions section in the revised manuscript. Please see page 18, line 723-733 in the revised manuscript.

Point 10: Legends should be self-explained, and some data is missing regarding number of independent experiments performed, indication of the experimental unit and better description of the statistics analysis/results (for each experiment), as mentioned.

Response 10: As per reviewer’s suggestion, we rewrote all the figure legends in the revised manuscript.

Point 11: I did not find a comparison/discussion/result integration between other similar works such as in J Pathol. 2018 May;245(1):85-100, that describes how liraglutide reverses cognitive impairment in mice and attenuates insulin receptor and synaptic pathology in a non-human primate model of Alzheimer's disease!

Response 11: We thank the reviewer for the valuable suggestion. We have reviewed the article and included it in the discussion section as reference number 46, of the revised manuscript, page 13, lines 456 & 466.

Response 12: All the graphs (Fig. 1-5) should indicate the individual data points to understand variability within each group. This also contributes to data transparency, robustness and data interpretation.

Response 12: We changed the bar graphs (Fig 1-5) to vertical point scatter plot graph to better interpret the variability within each group. These changes were included in the results section of the revised manuscript as Figures 2, 3 4, 5 & 6.

Response 13: A final cartoon/graphical abstract/scheme would be a plus to the paper

Response 13: We have added a cartoon abstract and uploaded it separately along with the revised manuscript.  

Round 2

Reviewer 1 Report

Point 1. By injecting STZ into 5xFAD mice the authors have created a ‘mixed’ SAD/FAD model which has demonstrably affected the results of this manuscript. The entire 5xFAD arm of the experiment should be removed as it does not contribute to the manuscript but only causes confusion.

Point 2. Ln 644 states “STZ was prepared fresh by dissolving 3 mg/kg of it into normal saline, which was injected via i.c.v. route.” Please correct this by providing the stock solution.

Point 8. It was not indicated anywhere in the original manuscript that a sample size of only 3 mice per group were used throughout analyses as the methods indicate that each group consisted of 6 mice (lines 603-607). The authors must provide sample size calculations to demonstrate that a sample of 3 mice per group were sufficient to power this study.

Point 9. These images still show nothing.

Point 10. Abeta is a 4kDa proteolytic product that can form oligomers. The authors have resolved their samples on 4-20% tris-glycine gels that has generally been shown to poorly resolve Abeta. The authors have identified species at 50-80 kDa as being oligomers of Abeta. The literature does not support this view. The authors have disregarded the most important Abeta species that is generally observed in these samples – low molecular weight species (4-16kDa). These samples must be re-run on either gradient tris-tricine or Bis-Tris gels in order before claims regarding Abeta can be made. Unless the authors can provide immuno-adsorption experiments that demonstrate that the 50-80kDa species are in fact Abeta, they need to quantify the known lower molecular weight species.

Point 11. The authors response regarding GAPDH signals are not correct. This reviewer has not reviewed previous publications from this group and cannot speak to their other publications. The comments regarding GAPDH were made based on the principles of western blot quantification. The authors response is not justified.

Point 12. Unless new instruments are used for each sample, the authors will be surprised by what they find if they perform the test suggested.

Point 13. The authors response further substantiates the suggestions made by this reviewer that the 5xFAD mice should be removed from this manuscript. If one is presenting these results then the study should be completed by performing the extra experiments and not indicating that it warrants further investigation as the study is incomplete.

Reviewer 3 Report

Authors did a good work in the revision of the manuscript. The manuscript is now more robust and transparent. They could solve many of the limitations and major revisions. The manuscript, meaning the results, data presentation and conclusion, is now more clear and with a more appropriate conclusion/discussion.